# Proteomic and Transcriptomic Analysis for Identification of Endosymbiotic Bacteria Associated with BYDV Transmission Efficiency by *Sitobion miscanthi*

**DOI:** 10.3390/plants11233352

**Published:** 2022-12-02

**Authors:** Wenjuan Yu, Emilie Bosquée, Jia Fan, Yong Liu, Claude Bragard, Frédéric Francis, Julian Chen

**Affiliations:** 1MOA Key Laboratory of Integrated Management of Pests on Crops in Southwest China, Institute of Plant Protection, Sichuan Academy of Agricultural Sciences, Chengdu 610066, China; 2Functional and Evolutionary Entomology, Gembloux Agro-Bio Tech, University of Liege, Passage des Déportés 2, 5030 Gembloux, Belgium; 3State Key Laboratory for Biology of Plant Disease and Insect Pests, Institute of Plant Protection, Chinese Academy of Agricultural Sciences, Beijing 100193, China; 4College of Plant Protection, Shandong Agricultural University, Tai’an 271018, China; 5Applied Microbiologye-Phytopathology, Earth and Life Institute, UCLouvain, Croix du Sud L7.05.03, 1348 Louvain-la-Neuve, Belgium

**Keywords:** *Sitobion miscanthi*, barley yellow dwarf virus, endosymbiont, transmission efficiency

## Abstract

*Sitobion miscanthi*, an important viral vector of barley yellow dwarf virus (BYDV), is also symbiotically associated with endosymbionts, but little is known about the interactions between endosymbionts, aphid and BYDV. Therefore, two aphids’ geographic populations, differing in their BYDV transmission efficiency, after characterizing their endosymbionts, were treated with antibiotics to investigate how changes in the composition of their endosymbiont population affected BYDV transmission efficiency. After antibiotic treatment, *Rickettsia* was eliminated from two geographic populations. BYDV transmission efficiency by STY geographic population dropped significantly, by −44.2% with ampicillin and −25.01% with rifampicin, but HDZ geographic population decreased by only 14.19% with ampicillin and 23.88% with rifampicin. Transcriptomic analysis showed that the number of DEGs related to the immune system, carbohydrate metabolism and lipid metabolism did increase in the STY rifampicin treatment, while replication and repair, glycan biosynthesis and metabolism increased in the STY ampicillin treatment. Proteomic analysis showed that the abundance of symbionin symL, nascent polypeptide−associated complex subunit alpha and proteasome differed significantly between the two geographic populations. We found that the endosymbionts can mediate vector viral transmission. They should therefore be included in investigations into aphid–virus interactions and plant disease epidemiology. Our findings should also help with the development of strategies to prevent virus transmission.

## 1. Introduction

Wheat, *Triticum aestivum* (L.), the third largest food crop in China, is severely attacked by aphids, including *Sitobion miscanthi* (Fabricus), one of the most economically important insect pests. This aphid directly pierces wheat plants and sucks the phloem sap, thus indirectly acting as the main vector for barley yellow dwarf virus (BYDV). BYDV is transmitted in a persistent and circulative pattern, causing wheat yellow dwarf virus disease. BYDV is a major phytovirus of the genus *Luteovirus* (family *Luteoviridae*), which can adversely affect almost all members of the *Gramineae*, causing severe crop losses worldwide [1]. 

Five strains of BYDV, based on their primary aphid vectors, have been identified [1]. Each strain is only transmitted efficiently by its corresponding aphid species [2]. A virus isolate can be transmitted with various efficiencies by different geographic populations of an aphid species, the same way an aphid geographic population can transmit different virus isolates also with different efficiencies [1,2].

Almost all aphids (Hemiptera: Aphididae) are closely associated with bacterial endosymbionts, all establishing a close relationship with their host aphid. Obligatory (or primary) endosymbionts, for example, *Buchnera aphidicola*, reside in the cytoplasm of aphid’s bacteriocytes, hypertrophied cells found in the abdomen, and synthesize essential amino acids and other nutrients that are lacking in the plant sap diet [3,4,5]. A number of aphids harbor several inherited facultative symbionts (or secondary, namely S−symbiont), which can be vertically transmitted at low levels by colonizing new host aphids. At least 10 S−symbionts have been detected in aphids [6,7]. These symbionts differ remarkably among the lineages in morphology, quantity and localization in the host insect [8]. 

Although the endosymbionts are not strictly required for host survival, they might provide a selective advantage in certain conditions [9]. However, little is known about the effect of the endosymbionts on the interaction between aphids and vectored viruses. Previous experiments showed that the geographic origin of aphids and the host plant species can affect the virus transmission [10,11] and that endosymbiont populations vary with the same two factors (geographic origin of aphid and host plat species) [9]. Thus, we hypothesized that a different composition of endosymbionts harbored by the aphids’ geographic population might be associated with the difference in their efficiency to transmit the virus. In the present study, first, we characterized the composition of endosymbiont flora of two geographic populations (STY and HDZ) of *S. miscanthi* treated with antibiotics. Second, we assessed their capacity for virus transmission with or without treatments. Third, we compared the differences in the genes and metabolic pathways in *S. miscanthi* across antibiotic treatments, and then screened the specially expressed proteins in the two geographic populations by proteomic approaches. Last, by combining the BYDV transmission, metabolic pathways and the specially expressed proteins, we attempted to analyze the role of endosymbiotic bacteria in the BYDV transmission process.

## 2. Results

### 2.1. Viral Transmission

The effect of the antibiotics (ampicillin and rifampicin) on vector transmission of BYDV−PAV isolates (CN and BE) was compared for each *S. miscanthi* geographic population (STY and HDZ) with the control (Table 1). The efficiencies of BYDV−PAV transmission by *S. miscanthi* tested were reduced when the aphids were previously treated with antibiotics. When the STY geographic population was infected with BYDV−PAV−CN isolate, the inhibition rates ranged from 25.0% to 44.2% after antibiotic treatment, corresponding to significant difference with antibiotic−free treatments (t = 7.93 and *p* < 0.001). For the HDZ geographic population, the inhibition rates ranged from 14.2% to 23.9%, corresponding to a significant reduction in the virus transmission rate (t = 4.37 and *p* < 0.001). The highest inhibition rate of virus transmission occurred in the STY geographic population treated with ampicillin. 

As to the Belgian virus isolate, BYDV−PAV−BE, the observed virus transmission inhibition rates were low after treatment with two antibiotics for the HDZ aphid geographic population (about 3.5%) and not significantly different from the control (t = 0.20; *p* = 0.842). For the STY geographic population, the inhibition rate was higher (from 21.4% to 25.84%) and similar to the one obtained with the Chinese virus isolate. The inhibition rates of BYDV−PAV−BE isolate transmission were significantly higher with STY than HDZ geographic populations whatever the considered antibiotic (t = 10.18 and t = 7.19 for both *p* < 0.001).

Whether aphids were infected with BYDV−PAV−CN or BYDV−PAV−BE isolates, the percentage of virus transmission of STY geographic population aphids that were treated with ampicillin was higher than aphids treated with rifampicin; in contrast, the percentage of virus transmission of HDZ geographic population aphids that were treated with ampicillin was lower than aphids treated with rifampicin.

### 2.2. Symbiotic Population Screening

As expected, the primary symbiont, *Buchnera aphidicola*, was detected in the two aphids’ geographic population. However, the composition of the S−symbionts differed in the two geographic populations. PASS1, PASS2, PAUS, *Rickettsia2* and *Wolbachia* were not detected in any sample. After antibiotic treatment, only some symbionts were eliminated from aphid geographic populations. *Rickettsia1* was eliminated from both the STY geographic population and the HDZ geographic population (Appendix A).

The relative abundance of the endosymbiont genes in the two aphids’ geographic populations with different treatments (ampicillin and rifampicin) was analyzed using a comparative ΔΔCt method (Figure 1). The abundance of endosymbiont was significantly higher than those aphids fed with antibiotics diets, except *Buchnera aphidicola* from HDZ clone was treated with rifampicin and *Spiroplasma* from STY clone treated with ampicillin. 

### 2.3. Transcriptome Overview

The transcriptomes of *S. miscanthi* feeding on free, antibiotic and BYDV−PAV were sequenced and compared. A total of 129.33 Gb of clean data was obtained from the 18 treatments, and each of these samples contained ≥ 5.4 Gb of data with Q30 quality scores ≥ 92.55% (Appendix A), and 56,196 unigenes were identified with 34,941 unigenes having annotation information (Appendix A).

The gene expression levels were used to conduct a PCA for each of the biological replicates. Each replicate from the same group was clustered closely together, which suggested that the repeatability of each treatment was satisfactory, and the samples from different antibiotics of *S. miscanthi* reared with BYDV were clustered far from each other and the control groups, which indicated that aphids feeding on antibiotics induced significant changes in gene expression (Figure 2A). The *p* value ≤ 0.01 (false discovery rate [FDR] adjusted) and Log2−fold change (Log2FC) ≥ 1 or ≤−1 were set as thresholds for DEGs in aphids at different treatments. Then, these identified DEGs were used for further analysis. Up− and downregulated DEGs were identified between different treatments, respectively (Figure 2B). The distributions of up− and downregulated genes were calculated for rifampicin or ampicillin and are presented in a Venn diagram (Figure 2C,D). 

GO analysis was used for the functional classification of the DEGs in aphids after rearing with antibiotics. The top 30 enriched GO terms of all DEGs are shown in Table 2. Among the STY−Free−vs.−STY−Vir, STY−Free−vs.−STY−Rif^+^−Vir and STY−Free−vs.−STY−Amp^+^−Vir, the top 10 upregulated DEGs, three genes (*CRC*, *CRA1*, adhesive plaque matrix protein−like) were annotated in three group, and one gene (*CRB*) was annotated in two antibiotic treatments. Among the top 10 downregulated DEGs, three genes (uncharacterized protein LOC100158873 precursor, RNA−binding protein 14, integumentary mucin C.1) were annotated in two antibiotic treatments.

Compared to the rifampicin–*S. miscanthi* and ampicillin–*S. miscanthi*, fed with BYDV for 48 h, rifampicin–*S. miscanthi* had more immune system−, lipid metabolism− and carbohydrate metabolism−related DEGs upregulated, but ampicillin–*S. miscanthi* had more replication and repaired−related and glycan biosynthesis and metabolism−related DEGs upregulated (Figure 3). 

### 2.4. Protein Identification

A proteomic work was conducted by 2D−DIGE to monitor the different protein expression from two geographic populations: STY geographic population and HDZ geographic population. More than 250 spots were generated but 86 proteins were selected for identification (Figure 4), mainly (66.0%) with homology with proteins from *Acyrthosiphon pisum* (which is actually the only aphid species for which the entire genome has been sequenced) (Figure 5), and classified into 12 functional categories based on their functions (Figure 6 and Table 3). 

From the variation in 86 proteins, only 14 proteins were upregulated for the inefficient vector against 63 proteins upregulated for the efficient vector. 

## 3. Discussion

For the biological function of an individual symbiont in such complex systems to be understood, a moderate rifampicin treatment of *A. pisum* and *S. miscanthi* has been shown to selectively eliminate *Buchnera aphidicola*, and ampicillin selectively eliminated *Regiella* and *Serratia* [12,13,14]. However, in this study, *Buchnera aphidicola* was found in all *S. miscanthi* geographic populations after treating with rifampicin, but its concentration was reduced. We speculate that rifampicin treatment might reduce symbiont density but not completely remove *Buchnera aphidicola*. When *S. miscanthi* was fed an ampicillin or rifampicin diet, *Rickettsia* was systematically eliminated in the present study; the *Rickettsia* symbiont, like other γ−proteobacteria symbionts identified in secondary mycetocytes and sheath cells from *A. pisum*, was more exposed to antibiotics and thus eliminated [15]. Many studies illustrated that PABS was localized not only in secondary mycetocytes and sheath cells, but also in the hemolymph [4,12], so its concentration was reduced by antibiotics. *Arsenophonus* and *Spiroplasma* were successfully eliminated after treatment with rifampicin, but not with ampicillin. This result is similar to a study on *Bemisia tabaci* where rifampicin inactivated a higher percentage of *Arsenophonus* than rifampicin [16]. 

As expected, virus transmission was reduced following the antibiotic treatment; the endosymbionts were presumably killed or inhibited, decreasing the efficiency of BYDV transmission. Since *Rickettsia* was the only S−symbiont in the HDZ geographic population, *Rickettsia* might be an important factor in the facilitation of BYDV transmission. Similarly, Kliot et al. [17] showed that a *B. tabaci* strain infected with *Rickettsia* acquired more tomato yellow leaf curl virus (TYLCY) from infected plants, retained the virus longer and exhibited nearly double the transmission efficiency than a non−infected strain that had the same genetic background. When TYLCV was acquired, it induced massive activation of gene expression in the *Rickettsia* uninfected population, whereas in the *Rickettsia*−infected population, the virus induced massive downregulation of gene expression. Fitness and choice experiments revealed that *Rickettsia*−infected whiteflies are always more attracted to TYLCV−infected plants [18]. When Sakurai et al. [15] investigated a *Rickettsia* symbiont using electron microscopy, virus−like particles were sometimes observed in association with *Rickettsia* cells. So, *Rickettsia* could play a crucial role in BYDV transmission. We applied the model that could calculate insect symbionts and insect vector contributions to pathogen transmission by insects, proposed by Patricia et al. [19], to test whether *Rickettsia* is involved in BYDV−PAV transmission. The fraction of the transmission efficiency provided by *Rickettsia* is equal to 0.14 (ampicillin) and 0.24 (rifampicin); these data indicate that *Rickettsia* contributes substantially to the BYDV−PAV transmission efficiency, but not as much as the insect vector contribution. In the HDZ geographic population, *Buchnera aphidicola* density was reduced by rifampicin, and *Rickettsia* was removed; rifampicin was more effective than ampicillin at reducing virus transmission, providing evidence that *Rickettsia* may act in concert with *Buchnera aphidicola* to influence the BYDV transmission of *S. miscanthi*.

The circulative transmission pathway through an aphid vector involves complex interactions between viral proteins and vector−associated compounds [8]. Using the proteomic and transcriptomic analysis, we identified differentially expressed proteins of the *S. miscanthi* STY geographic population. 

### 3.1. Cell Signaling

The proteasome is a protein−destroying apparatus involved in many essential cellular functions. The 26S proteasome is a large, multi−subunit proteolytic machine found in the nucleus and cytoplasm of mammalian cells. It comprises a 20S cylindrical catalytic core and two 19S regulatory caps. The 20S core contains four heptameric rings, two of which contain seven alpha subunits and two that contain seven bate subunits [20]. The proteasome, protein ubiquitination machinery or both (Ubiquitin/26S proteasome (UPS) pathway) are the major types of proteolytic machinery found in eukaryotes and are associated with immune responses to pathogen invasion, linked to the activation and subcellular localization of virus replication or movement protein complexes [21]. The turnip yellow mosaic virus (TYMV) movement protein is degraded by the proteasome; UPS regulates the accumulation of TYMV during viral infection and therefore decreases viral replication [22]. UPS could protect against viral infection by regulating the proliferation and transport of viruses in host cells via targeting the degradation of many viral proteins [21]. *Laodelphax striatellus* 26S proteasome played a defensive role against RBSDV infection by regulating RBSDV accumulation [23]. The proteasome of *R. padi* is strongly implicated as an antiviral immune response against the movement process of BYDV−GPV in the body of its aphid vectors [24]. We found that most proteasomes were upregulated in highly BYDV−PAV transmission−efficient vectors; we inferred that the proteasome may enhance the BYDV−PAV transmission efficiency in *S. miscanthi*.

### 3.2. Membrane Transport

The nascent polypeptide−associated complex (NAC) is a key regulator of proteostasis to provide the cell with a regulatory feedback mechanism in which translational activity is also controlled by the folding state of the cellular proteome and the cellular response to stress [25]. The alpha subunit is one of two subunits (alpha and beta subunit) of the NAC and contributes to the prevention of inappropriate interactions. The NAC subunit alpha of *Sogatella furcifera*, which strongly interacted with southern rice black−streaked dwarf virus, is a major outer capsid protein [26]. The relative strengths of the interactions between the BYDV−GPV CP and NAC subunit alpha were greater than the negative control [24]. The NAC domain protein was originally characterized as the first ribosome−associated protein to contact the emerging viral polypeptide chain. Liu et al. [27] found that the NAC of small brown planthopper was confirmed in an interaction with *rice stripe virus* (RSV) nucleocapsid (pc3), and they proposed that NAC binding to RSV pc3 may play an important role in viral replication. The NAC domain protein can also enhance replication of tomato leaf curl virus by binding the viral replication accessory protein [28]. The NAC subunit alpha was upregulated in the STY geographic population, so the NAC subunit alpha perhaps binds with BYDV and plays an important role in viral replication. 

### 3.3. Stress Tolerance

Another well−known protein family related to various stress responses varying between the two geographic populations was that of heat shock proteins (Hsps). In citrus tristeza virus (CTV), the protein P65 (the homologue of Hsp70) was essential for virion assembly and acted to restrict encapsidation by the minor coat protein to the 5′ end of the virion [29], and P65 was found have a role in the aphid transmission of the CTV process [30]. The members of the Hsp70 family were upregulated in the STY geographic population; thus, we hypothesize that Hsp70 may be involved in the aphid transmission of BYDV.

Symbionin is abundantly synthesized by endosymbiotic bacteria *Buchnera aphidicola* harbored in the bacteriocyte cells and is unlikely to be exported into the aphid hemolymph [31]. Symbionin−like molecules are found in major aphid species (including BYDV vectors), except those belonging to *Phylloxeridae* [24]. The interaction of a coat protein–read−through protein with symbionin was considered an essential factor to stabilize virions in the hostile environment of the aphid hemolymph. Symbionin has been shown to bind to purified luteoviruses in vitro or to a recombinant luteovirus read−through polypeptide [32,33,34,35]. However, the interaction’s contribution to transmission is controversial because luteoviruses bind symbionins of both vector and non−vector aphids [35], and recent studies on localization in vivo of the chaperone question its availability for interaction [36,37]. When aphids were cured of endosymbionts by treatment with antibiotics, their ability to transmit the virus was significantly reduced and the amount of coat protein was diminished. Strangely, the amount of read−through protein was not affected [32,33]. After the aphids were treated with rifampicin, the BYDV−PAV transmission efficiency was decreased by a quarter or so. The results of these experiments must be interpreted carefully—the destruction of the endosymbionts is likely to have dramatic effects on the metabolism and physiology of the aphids, and these changes may be directly or indirectly responsible for the effects on luteovirus protein detection and virus transmission. So, we propose that *Buchnera aphidicola* is involved in virus movement within the aphids, but we do not specify whether the effect of *Buchnera aphidicola* on transmitting viruses is direct or indirect. 

### 3.4. Immune System

Insects rely on their immune system to fight against pathogens [38]. As shown in our results, whether aphids feed with or without antibiotics, after feeding on BYDV−PAV, the DEGs related to immunity in S. miscanthi were upregulated, including the MAPK signaling pathway, lysosomes, antigen processing and presentation, ubiquitin−mediated proteolysis, insect hormone biosynthesis and peroxisomes [39,40]. These results suggest that decreased bacteria *Buchnera aphidicola* has more of an effect on the immune system than secondary endosymbiont. The proteins involved in the cytoskeleton were also differentially expressed, which may be related to the immune response [41]. There have been previous studies showing that viruses can interact with and reorganize host cytoskeleton components for intercellular trafficking and infection processes [42]. In addition, the cytoskeleton is also commonly involved in the intracellular transport of viruses [43,44,45]. 

Similarly, the two geographic populations of *S. miscanthi* were collected from different regions, which differed in the prevalence of wheat yellow dwarf disease. STY was from northwestern China where BYDV disease is severe; HDZ was collected from the Huang−Huai region of China, where BYDV disease is less severe [46]. On the other hand, the STY geographic population has a higher diversity of symbionts than HDZ does, which suggests that the aphid’s viral transmission efficiency results from increased fitness to different levels of stress posed by BYDV in the wheat−growing areas and that the symbionts may mediate the evolution of aphid fitness. Such speculation awaits further experimental evidence. 

## 4. Conclusions

Whether *Buchnera aphidicola* density was reduced or S−symbiont was removed, BYDV transmission efficiencies of *S. miscanthi* were all reduced, results which suggest that endosymbiotic bacteria take part in BYDV transmission. When only S−symbiont *Rickettrsia* was removed, BYDV transmission was reduced significantly, suggesting Rickettsia could play a crucial role in BYDV transmission, but the function of the other S−symbionts needs deeper research. Upon further analysis, we found that the number of DEGs related to the immune system, carbohydrate metabolism and lipid metabolism were increased when *Buchnera aphidicola* density was reduced, but replication and repair, glycan biosynthesis and metabolism were increased when S−symbionts were eliminated. This result will contribute to further studies on exploring the immune response of *S. miscanthi* to viruses. As the reports on endosymbionts mediating the interaction of vector and virus transmission are scarce, our research may provide insight into the relationship between endosymbiont and *luteovirus* transmission. Work on virus transmission efficiencies of aphids as affected by endosymbionts should be promoted to better understand the pathway of the virus in the aphid and to develop new tools to prevent virus transmission. Indeed, identification of molecular receptors in aphids should help discover competitors that prevent binding of the virus and reduce viral transmission.

## 5. Materials and Methods

### 5.1. Aphids and Virus

Two *S. miscanthi* geographic populations were collected from winter wheat fields in Taiyuan−Shanxi Province (STY) and Dengzhou−Henan Province (HDZ). These two geographic populations were selected from a previous study [10] in which STY was the aphid geographic population that was the most efficient for the transmission of BYDV, contrary to the HDZ geographic population, which had very low efficiency. So that the risk of collecting the same genotype in multiple sampling times was reduced, individual aphids were collected from plants growing at least 10 m apart.

Two geographic populations were reared separately on potted seedlings of wheat *cv. Toison d’Or* (susceptible to aphids) in the second leaf stage. Each pot was isolated in a transparent, plastic, ventilated, cylindrical cage (10 × 30 cm) covered with gauze on the top. Aphids and plants were maintained in a greenhouse compartment (22 ± 1 °C, 60 ± 5% RH and 16:8 h l:d).

BYDV−PAV−BE (Louvain-la-Neuve, Belgium) and BYDV−PAV−CN (Yangling, Shaanxi Province, China) isolates were separately maintained on seedlings of wheat *cv. Toison d’Or* infested with *S. miscanthi* in a greenhouse compartment (20 ± 1 °C, 60 ± 5% RH and 16:8 h l:d) [10].

### 5.2. Antibiotic Treatment and Viral Transmission

To selectively eliminate *Buchnera aphidicola* or S−symbiotic, first−instar (or 24 h old) nymphs of the two geographic populations (STY and HDZ) were fed an artificial diet (15% *w*/*v* sucrose solution with and without 50 μg mL^−1^ rifampicin or ampicillin (Sigma, St. Louis, MO, USA)) confined between two stretched Parafilm^®^ membranes on an opaque cylinder for 48 h. Aphids were then transferred to the typical virus−acquisition diet (BYDV−infected wheat tissue grinded in a 15% *w*/*v* sucrose solution) for 48 h of virus acquisition. After acquiring the virus, aphids were transferred to a 7−day−old healthy wheat seedling (one aphid per test plant) protected by a plastic cage on the pot. After a 5−day inoculation access period, aphids were removed and plants were grown for 15 days in a greenhouse before testing the presence of the virus by DAS−ELISA according to the manufacturer’s instructions (DSMZ, Braunschweig, Germany). The artificial diet without antibiotics (“antibiotic−free”) was used as a control. Fifty wheats were formed for one biological sample; three biological replicates were performed for each treatment.

The inhibition rate of virus transmission was calculated as: ((Transmission efficiency for treated samples − Transmission efficiency for control samples)/Transmission efficiency for controls) × 100.

### 5.3. DNA Extraction

Aphids were soaked with 70% ethanol and sterile water several times to remove bacteria from their surface. Total DNA was extracted from 50 aphids of each *S. miscanthi* geographic population (STY and HDZ) following the manufacturer’s instructions (DNeasy Tissue Kit, QIAGEN, Frankfurt, Germany). The quantity and purity of extracted DNA were evaluated using a spectrophotometer NanoDrop 1000 (Thermo Fisher Scientific, Pittsburgh, PA, USA). Samples were then diluted to 500 ng μL^−1^.

### 5.4. Symbiotic Population Screening

To identify respective endosymbiotic bacteria, DNA from the samples was amplified using the specific primers of Tsuchida et al. [47] and Fukatsu et al. [48]. Amplifications were performed in a reaction volume of 20 μL including 2 μL DNA, 10 μL 2 × *Taq* PCR MasterMix (Invitrogen, Carlsbad, CA, USA), 1 μL forward primer (10 mM), 1 μL reverse primer (10 mM) and 6 μL ddH_2_O. The PCR cycling conditions were as follows: 95 °C for 4 min, 40 cycles at 95 °C for 30 s, 55 °C for 30 s, 72 °C for 30 s and final extension at 72 °C for 5 min. The amplified product was separated in 2% agarose gel and stained with ethidium bromide (Thermo Scientific, Waltham, MA, USA).

The relative abundance of *Buchnera aphidicola* and S−symbiont before/after antibiotic control was assessed using quantitative real−time PCR (qPCR). Specific primer pairs for qPCR were designed with Primer 3 (Appendix A), and qPCR was performed on an ABI 7500 Real−Time PCR System (Applied Biosystems, Carlsbad, CA, USA). The reference gene, NADH dehydrogenase, was used for normalizing target gene expression and correcting for sample−to−sample variation. The qPCR reactions were performed in 20 μL reactions containing 2 μL of sample DNA, 10 μL of SYBR Premix Ex Taq (TaKaRa, Beijing, China), 0.5 μL of each primer (10 μM), 0.4 μL of Rox Reference Dye and 6.6 μL of sterilized H_2_O. The qPCR cycling parameters were 95 °C for 30 s, followed by 40 cycles of 95 °C for 15 s and 60 °C for 30 s. Next, the PCR products were heated to 95 °C for 15 s, cooled to 60 °C for 1 min and 95 °C for 15 s to measure the dissociation curves. qPCR reaction for each sample was carried out with three technical replicates and three biological replicates. Standard curves for reference genes and candidate genes were generated by gradient dilution to identify proper primers with 95–110% amplification efficiency and without nonspecific amplification. The relative abundance of aphid endosymbiont was normalized to the aphid housekeeping gene NADH and calculated using the comparative Ct method according to Vandesompele’s method (2^−ΔΔCt^) (2002) [49].

### 5.5. RNA Extraction, Library Construction, and RNA Sequencing

The first−instar nymphs of STY geographic population *S. miscanthi* were reared on 15% *w*/*v* sucrose solution, 50 μg mL^−1^ rifampicin or ampicillin for 48 h, then one part of aphids transferred to feed with BYTV for 48 h. For each treatment (STY−free, STY−Vir, STY−Rif^+^, STY−Amp^+^, STY−Rif^+^−Vir, STY−Amp^+^−Vir), three experimental replicates were used. For each replicate sampling, 30 individual aphids were collected and then flash−frozen using liquid nitrogen and stored at −80 °C. Total RNA was extracted using a Trizol reagent kit (Invitrogen, Carlsbad, CA, USA) according to the manufacturer’s protocol. RNA quality was assessed on an Agilent 2100 Bioanalyzer (Agilent Technologies, Palo Alto, CA, USA) and checked using RNase free agarose gel electrophoresis. After total RNA was extracted, eukaryotic mRNA was enriched by Oligo(dT) beads, while prokaryotic mRNA was enriched by removing rRNA by Ribo−ZeroTM Magnetic Kit (Epicentre, Madison, WI, USA). Then, the enriched mRNA was fragmented into short fragments using fragmentation buffer and reverse−transcribed into cDNA with random primers. Second−strand cDNA was synthesized by DNA polymerase I, RNase H, dNTP and buffer. Then, the cDNA fragments were purified with the QiaQuick PCR extraction kit (Qiagen, Venlo, The Netherlands), end−repaired, A base−added and ligated to Illumina sequencing adapters. The ligation products were size−selected by agarose gel electrophoresis, PCR−amplified and sequenced using Illumina novaseq 6000 by Gene Denovo Biotechnology Co. (Guangzhou, China). 

### 5.6. RNA−Seq Data Analysis

To obtain high−quality reads, the reads containing adaptor sequences, more than 10% of unknown nucleotides (N), and low−quality (Q−value ≤ 20) bases were removed [50]. Transcriptome de novo assembly was carried out with the short reads assembling program Trinity [51]. The unigene expression was calculated and normalized to RPKM (reads per kb per million reads) [52]. Principal component analysis (PCA) was performed with R package models (http://www.r-project.org/) accessed on 10 February 2022 in this experience. RNA differential expression analysis was performed by DESeq2 [53] software between two different groups (and by edgeR (6) between two samples). The genes with a false discovery rate (FDR) below 0.05 and absolute fold change ≥ 2 were considered differentially expressed genes. Basic annotation of unigenes includes protein functional annotation, pathway annotation, COG/KOG functional annotation and Gene Ontology (GO) annotation. To annotate the unigenes, we used BLASTx program (http://www.ncbi.nlm.nih.gov/BLAST/, accessed on 10 February 2022) with an E−value threshold of 1 × 10^−5^ to the NCBI non−redundant protein (Nr) database (http://www.ncbi.nlm.nih.gov, accessed on 10 February 2022), the Swiss−Prot protein database (http://www.expasy.ch/sprot, accessed on 10 February 2022), the Kyoto Encyclopedia of Genes and Genomes (KEGG) database (http://www.genome.jp/kegg, accessed on 10 February 2022) and the COG/KOG database (http://www.ncbi.nlm.nih.gov/COG) on 10 February 2022. Protein functional annotations could then be obtained according to the best alignment results. 

### 5.7. Sample Preparation for 2−D DIGE

Fresh aphids (20 mg) collected from stocks of the HDZ geographic population or STY geographic population after feeding on the BYDV−free wheat seedlings were grinded in 100 μL UT buffer (7M Urea, 2M Thiourea, 0.5% (*w*/*v*) CHAPS) and centrifuged at 15,000× *g* at 4 °C for 15 min. Proteins were extracted from collected supernatants using a 2D−Clean−up Kit according to the manufacturer’s instructions (GE Healthcare, Freiburg, Germany) and then resuspended in 50 μL rehydration buffer (6M Urea, 2M Thiourea, 10% (*w*/*v*) CHAPS, 1% (*w*/*v*) ASB14 and 30M Tris pH 8.5). The precipitated proteins were quantified using the RC−DC Microfuge Tube Assay (Bio−Rad, Hercules, CA, USA).

The protein extracts (25 μg) were labeled with cyanine dye (Cy2, Cy3, Cy5) following the standard protocol (Lumiprobe, Hannover, Germany). Before labeling, the pH of samples was adjusted to 8.5 with NaOH (100 mM). Two samples (STY or HDZ) labeled either with Cy3 or Cy5 were mixed with an internal reference standard protein mixture (which was pooled from 12.5 μg STY and 12.5 μg HDZ) labeled with Cy2. A conventional dye swap for DIGE was performed by labeling two replicates from each treatment group with one dye (Cy3 or Cy5) and the third replicate with the other two cyanine dyes. A non−labeled 500 μg sample of aphid protein mixture was added on the preparative gel for protein picking. Each mix of labeled proteins was diluted in UT−Tris buffer to obtain a volume of 225 μL. This volume was then adjusted to 450 μL with 225 μL IPG/DTT (4 µL 100× BioLyte^®^ 3/10 Ampholyte (Bio−Rad), 2 mg DTT (Sigma Aldrich) and 219 µL UT buffer).

### 5.8. 2−D DIGE and Gel Analysis

The mix of labeled samples was deposited on 24 cm ReadyStrip™ IPG Strips pH 3–10 NL (Bio−Rad) for the first−dimensional isoelectric focusing (IEF) (Protean^®^ i12 IEF Cell, Bio−Rad) for 9 h at 50 V and 15 °C. Then, the IEF was carried out at 200 V for 2 h, 10,000 V for 1 h and 10,000 for 4 h 30 min. In an IEF unit, the current was settled at 50 μA/strip.

Before starting the second−dimensional electrophoresis, strips were reduced for 15 min in a buffer containing 30% (*w*/*v*) urea, 83% (*v*/*v*) equilibration buffer and 0.83% (*w*/*v*) dithiothreitol (DTT), and then for a further 15 min in the same buffer but in which DTT was replaced with 2% (*w*/*v*) iodoacetamide (IAA). Strips were laid down on 2D HPE^TM^ Large Gels NF 12.5% acrylamide (Serva Electrophoresis GmbH, Heidelberg, Germany) and the second−dimensional electrophoresis was performed with the HPE FlatTop Tower (Serva) according to the manufacturer’s instructions. Then, the preparative gel was placed overnight in a fixation buffer (10% acetic acid, 30% ethanol and 60% H_2_O) and stirred. The scan of gels was performed at wavelengths corresponding to each cyanine dye with a Typhoon Ettan DIGE Imager (GE Healthcare, Freiburg, Germany). Gel images were analyzed using Nonlinear Progenesis Samespots (Nonlinear Dynamics, Newcastle Upon Tyne, United Kingdom), and protein spots were excised from the gel using an Ettan spotpicker robot (GE Healthcare). Selected gel pieces were processed as described by Bauwens et al., 2013 [54]. 

### 5.9. Protein Identification

Protein identification was possible thanks to the NCBI Database (restricted to Arthropoda) and a homemade aphid symbiont database. Searches were treated on the Mascot server 2.2.06 with BioTools^TM^3.2 (Bruker Daltonics). Proteins were retained only when their score was at least 45 and matched at least four peptides with error values < 100 ppm. The identified proteins were categorized according to metabolic function using the Kegg pathway database (http://www.genome.jp/kegg/pathway.html, accessed on 10 February 2022) and Expasy Proteomic tools (http://www.expasy.org/tools/, accessed on 10 February 2022), particularly the Biochemical–Metabolic pathway sections on 10 February 2022.

### 5.10. Statistical Analysis

For the viral transmission, an analysis of variance (ANOVA) was performed on the percentage of virus transmission of infected plants in different treatments using the GLM procedure in the SAS 9.1 program. Data were analyzed with Student’s *t*−test. For the qPCR, differences in transcript expression of same endosymbiont among different treatments were statistically analyzed with a one−way ANOVA using SAS 9.1 followed by Duncan’s Multiple Range Test. Differences in transcript expression of same endosymbiont with the same treatment between STY geographic population and HDZ geographic population were analyzed with Student’s *t*−test.

Quantitative differences in spot intensity among the two groups were analyzed by analysis of variance implemented in SAMESPOT, version 3.5. Differential regulation of proteins was compared by a log_2_−fold change approach. A Pearson’s chi−squared independence test implemented in R software (R−Core−Team, 2014) was used to test the association between groups (STY and HDZ geographic populations) and protein regulation (up− and downregulation). A heatmap was elaborated using Excel (Microsoft Corp., Redmond, Washington, DC, USA) to visualize proteins displaying increased and decreased expression.

## Figures and Tables

**Figure 1 plants-11-03352-f001:**
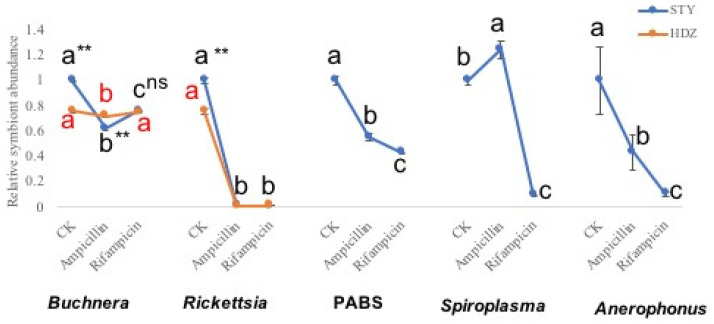
Relative abundance of endosymbiont in the Shanxi Taiyuan (STY) geographic population and Hennan Dengzhou (HDZ) geographic population of *Sitobion miscanthi* with and without antibiotic treatment. Comparison between the two aphids’ geographic population in one endosymbiont; ** significantly different (Student’s *t*−test, *p* < 0.01). Comparison among the expression profiles of endosymbiont treated with different antibiotic in the same aphid geographic population; “abc” significantly different (*p* < 0.01); “ns” no significantly different (*p* < 0.01).

**Figure 2 plants-11-03352-f002:**
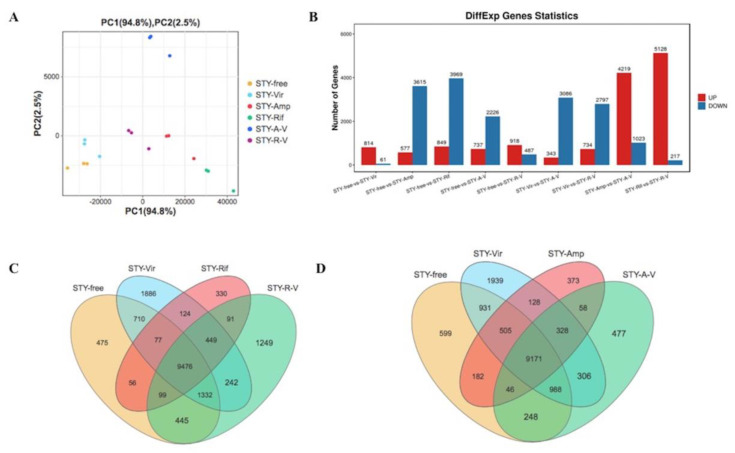
Transcriptomic overview of *Sitobion miscanthi* feeding on antibiotics and BYDV−PAV. (**A**) PCA plot of global transcriptome profiles. (**B**) Total number of transcripts that were significantly up− or down−regulated in response to aphids feeding on antibiotics and BYDV−PAV. (**C**) Venn diagram illustrating the number of genes up− or down−regulated by aphids feeding on rifampicin over the time course. *p* < 0.01 FDR and Log2 FC ≥ 1 or ≤−1. (**D**) Venn diagram illustrating the number of genes up− or down−regulated by aphids feeding on ampicillin over the time course. *p* < 0.01 FDR and Log2 FC ≥ 1 or ≤−1.

**Figure 3 plants-11-03352-f003:**
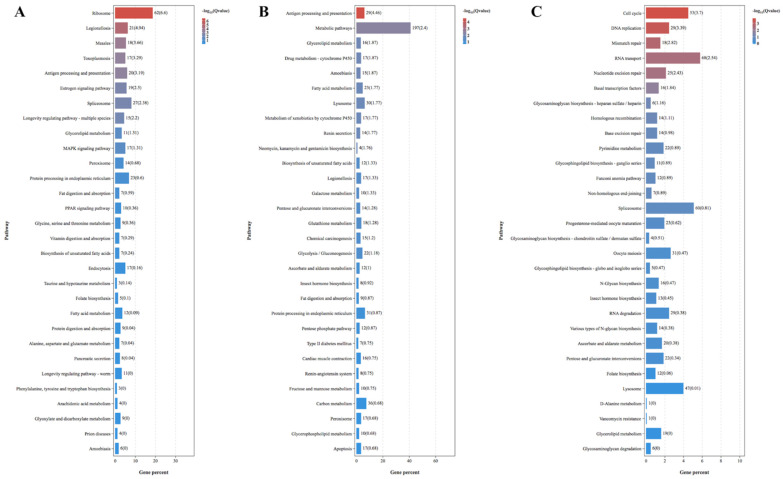
Pathway represents KEGG analysis of the differentially expressed genes (DEGs) in response to *Sitobion miscanthi* feeding on antibiotics and BYDV−PAV. (**A**) Top 30 pathway represents KEGG analysis of STY−Free−vs.−STY−Vir. (**B**) Top 30 pathway represents KEGG analysis of STY−Free−vs.−STY−Rif^+^−Vir. (**C**) Top 30 pathway represents KEGG analysis of STY−Free−vs.−STY−Amp^+^−Vir.

**Figure 4 plants-11-03352-f004:**
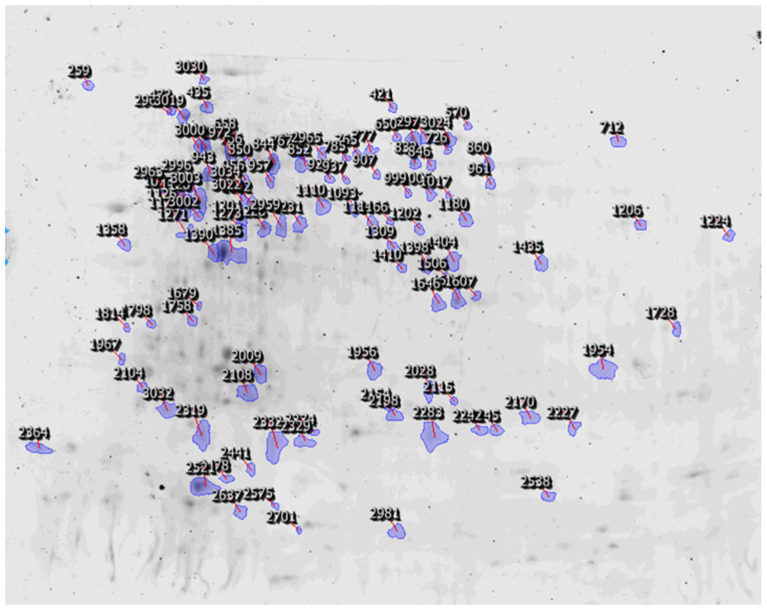
The 2D−DIGE gel separations of proteins from the STY geographic population and HDZ geographic population of *Sitobion miscanthi*.

**Figure 5 plants-11-03352-f005:**
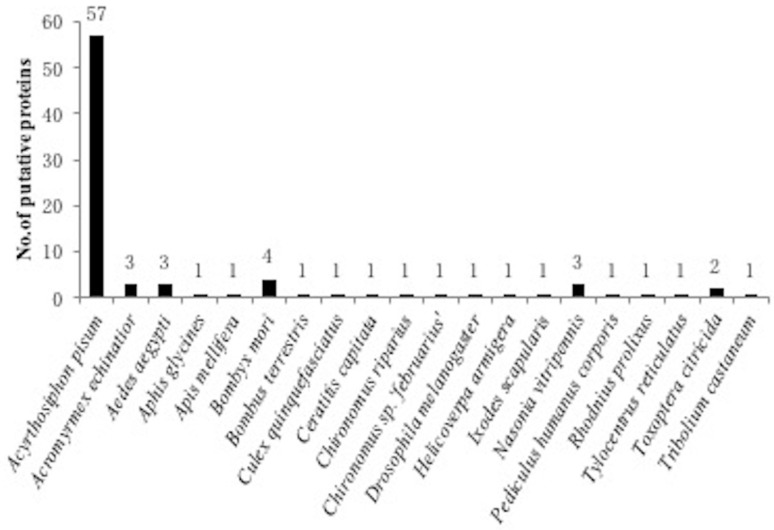
Distribution of the 86 putative proteins sequences similar to those of *Sitobion miscanthi* identified from other insect species in a BLASTX search.

**Figure 6 plants-11-03352-f006:**
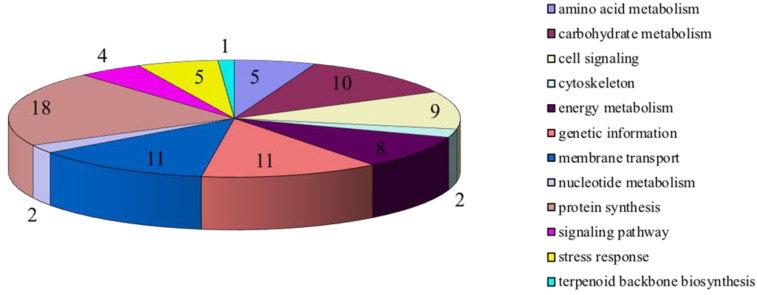
Pathway analysis of protein identified by 2D−DIGE gel separations from the STY geographic population and HDZ geographic population of *Sitobion miscanthi*.

**Table 1 plants-11-03352-t001:** Effect of antibiotic treatments on barley yellow dwarf virus (BYDV) transmission rates by two *Sitobion miscanthi* geographic populations.

Aphid Geographic Population	Virus Strain	Control ^a^ (%)	Inhibition Rate after Ampicillin (%) ^b^	Inhibition Rate after Rifampicin (%)	Statistics
Shanxi Taiyuan (STY)	BYDV−CN	56.14 (50/50 ^c^)	−44.20 ± 3.83 (37/50)	−25.01 ± 14.29 (47/50)	*t* = 7.935 **; df = 82; *p* < 0.0001
Hennan Dengzhou (HDZ)	BYDV−CN	24.57 (50/50)	−14.19 ± 9.55 (49/50)	−23.88 ± 3.65 (20/50)	*t* = 4.370 **; df = 59; *p* < 0.0001
Statistics			*t* = −17.858 **; df = 76; *p* < 0.0001	*t* = −0.349; df = 65; *p* < 0.7282	
Shanxi Taiyuan (STY)	BYDV−BE	60.95 (50/50)	−25.84 ± 10.64 (50/50)	−21.44 ± 12.97 (50/50)	*t* = 1.786; df = 98; *p* < 0.0772
Hennan Dengzhou (HDZ)	BYDV−BE	25.75 (50/50)	−3.450 ± 10.56 (45/50)	−3.896 ± 11.11 (49/50)	*t* = −0.199; df = 92; *p* < 0.8425
Statistics			*t* = 10.183 **; df = 93; *p* < 0.0001	*t* = 7.189 **; df = 97; *p* < 0.0001	

^a^ Control: BYDV transmission efficiency by *S. miscanthi* fed on BYDV sucrose without antibiotic. ^b^ Inhibition rate after ampicillin (%): aphids’ geographic population treated with 50 μg mL^−1^ ampicillin for 48 h, the BYDV transmission was inhibited. The inhibition rate of virus transmission = (transmission efficiency in treatment—transmission efficiency in control)/transmission efficiency of control × 100. Horizontal: Comparison between the two antibiotics in one aphid geographic population; ** significantly different (Student’s *t*−test, *p* < 0.01). Vertical: Comparison between the two aphids’ geographic populations treated with the same antibiotic; ** significantly different (Student’s *t*−test, *p* < 0.01). ^c^ No. of viruliferous plants/no. of tested plants.

**Table 2 plants-11-03352-t002:** Top 10 upregulated and downregulated DEGs in STY geographic population fed with/without antibiotics before *Sitobion miscanthi* was injected with BYDV−PAV.

Gene ID	log2(fc)	*p* Value	FDR	Symbol	Description
STY−Free−vs.−STY−Vir					
Upregulated					
Unigene0018501	15.9440	7.66 × 10^−4^	2.09 × 10^−2^	CRC	−
Unigene0037507	15.4923	1.34 × 10^−3^	3.30 × 10^−2^	CRA1	−
Unigene0053319	14.8411	4.01 × 10^−17^	1.48 × 10^−14^	−−	adhesive plaque matrix protein−like
Unigene0017731	14.0592	1.68 × 10^−13^	3.99 × 10^−11^	−−	uncharacterized protein FWK35_00010809
Unigene0032087	13.9178	3.30 × 10^−13^	7.59 × 10^−11^	−−	−
Unigene0048721	13.5507	5.94 × 10^−16^	1.83 × 10^−13^	−−	uncharacterized protein LOC113553374
Unigene0016176	13.3723	7.59 × 10^−228^	1.73 × 10^−223^	ORF2	−
Unigene0005546	7.7388	2.68 × 10^−19^	1.17 × 10^−16^	EbpIII	chemosensory protein CSP2
Unigene0000098	7.6686	1.74 × 10^−19^	7.79 × 10^−17^	Hsp68	heat shock protein 70 A1−like
Unigene0001872	7.3215	2.33 × 10^−9^	3.11 × 10^−7^	−−	alpha−tocopherol transfer protein
Downregulated					
Unigene0037317	−2.0849	2.36 × 10^−4^	8.28 × 10^−3^	−−	−
Unigene0020399	−1.3726	1.53 × 10^−23^	9.71 × 10^−21^	SNRPG	probable small nuclear ribonucleoprotein G
Unigene0044576	−1.3188	1.93 × 10^−4^	7.05 × 10^−3^	−−	hypothetical protein CINCED_3A023044
Unigene0055907	−1.2902	5.26 × 10^−6^	3.32 × 10^−4^	aurka−a	hypothetical protein AGLY_005943
Unigene0006746	−1.2896	5.74 × 10^−4^	1.66 × 10^−2^	−−	titin−like
Unigene0020224	−1.2634	1.71 × 10^−7^	1.59 × 10^−5^	LSM4	U6 snRNA−associated Sm−like protein LSm4
Unigene0026159	−1.2604	1.82 × 10^−8^	2.07 × 10^−6^	MAD2L1	mitotic spindle assembly checkpoint protein MAD2A
Unigene0024678	−1.2418	1.83 × 10^−31^	1.90 × 10^−28^	PCNA	proliferating cell nuclear antigen
Unigene0047899	−1.2065	1.88 × 10^−16^	6.22 × 10^−14^	−−	macrophage migration inhibitory factor−like
Unigene0029379	−1.1996	1.24 × 10^−21^	6.85 × 10^−19^	−−	leucine−rich repeat extensin−like protein 5
STY−Free−vs.−STY−Rif^+^−Vir					
Upregulated					
Unigene0018501	15.5264	1.27 × 10^−3^	1.64 × 10^−2^	CRC	−
Unigene0037141	15.3389	3.79 × 10^−20^	5.56 × 10^−18^	−−	uncharacterized protein LOC111030390
Unigene0037507	15.0004	2.32 × 10^−3^	2.72 × 10^−2^	CRA1	−
Unigene0055769	14.7281	2.33 × 10^−19^	3.10 × 10^−17^	−−	uncharacterized protein LOC111026481
Unigene0000110	14.6062	4.35 × 10^−19^	5.57 × 10^−17^	−−	uncharacterized protein LOC111039417
Unigene0024156	14.5291	1.51 × 10^−17^	1.67 × 10^−15^	−−	uncharacterized protein LOC111028874
Unigene0053319	14.3292	1.54 × 10^−15^	1.44 × 10^−13^	−−	adhesive plaque matrix protein−like
Unigene0043470	14.2754	4.11 × 10^−3^	4.37 × 10^−2^	CRB	−
Unigene0023887	14.2015	2.44 × 10^−20^	3.63 × 10^−18^	−−	uncharacterized protein LOC111038291
Unigene0012963	13.9784	1.55 × 10^−15^	1.45 × 10^−13^	EbpIII	ejaculatory bulb−specific protein 3−like
Downregulated					
Unigene0026858	−13.5407	3.33 × 10^−4^	5.17 × 10^−3^	−−	integumentary mucin C.1
Unigene0010013	−10.3420	1.82 × 10^−3^	2.22 × 10^−2^	−−	uncharacterized protein LOC100158873 precursor
Unigene0000011	−9.6662	1.82 × 10^−6^	4.96 × 10^−5^	−−	skin secretory protein xP2−like
Unigene0007325	−9.6555	2.53 × 10^−3^	2.92 × 10^−2^	SERPINB1	leukocyte elastase inhibitor
Unigene0005236	−9.6457	2.71 × 10^−3^	3.10 × 10^−2^	−−	uncharacterized protein LOC100163734 precursor
Unigene0043513	−9.5283	3.32 × 10^−4^	5.17 × 10^−3^	Tctp	translationally controlled tumor protein homolog
Unigene0012804	−9.5126	7.88 × 10^−5^	1.46 × 10^−3^	−−	calphotin
Unigene0020485	−9.4275	3.46 × 10^−3^	3.79 × 10^−2^	UQCRFS1	rieske iron−sulfur protein
Unigene0034136	−9.2576	1.70 × 10^−12^	1.13 × 10^−10^	−−	RNA−binding protein 14
Unigene0028055	−9.0465	1.15 × 10^−11^	6.90 × 10^−10^	Lypla1	acyl−protein thioesterase 1,2−like
STY−Free−vs.−STY−Amp^+^−Vir					
Upregulated					
Unigene0018501	15.7378	7.32 × 10^−4^	8.07 × 10^−2^	CRC	−
Unigene0037507	15.2877	1.28 × 10^−3^	1.31 × 10^−2^	CRA1	−
Unigene0010126	15.1055	4.02 × 10^−3^	3.46 × 10^−2^	RBCS	chloroplast ribulose−1,5−bisphosphate carboxylase/oxygenase small subunit 1
Unigene0048052	14.8538	4.85 × 10^−3^	4.04 × 10^−2^	AT2S2	−
Unigene0043470	14.4454	2.59 × 10^−3^	2.40 × 10^−2^	CRB	−
Unigene0025659	7.9282	0.00	0.00	−−	A−kinase anchor protein 14−like
Unigene0054542	7.8614	0.00	0.00	ACP21	cuticle protein 7−like
Unigene0002263	5.8154	0.00	0.00	−−	cuticle protein 64−like
Unigene0009391	5.6836	7.88 × 10^−9^	2.14 × 10^−7^	MT−CO1	cytochrome c oxidase subunit I
Unigene0039375	5.6716	5.42 × 10^−21^	4.77 × 10^−19^	Edg84A	larval cuticle protein A3A−like
Downregulated					
Unigene0034136	−15.4540	2.98 × 10^−4^	3.59 × 10^−3^	−−	RNA−binding protein 14
Unigene0052117	−14.3418	9.62 × 10^−4^	1.02 × 10^−2^	−−	uncharacterized protein LOC100166901
Unigene0026949	−13.8672	9.41 × 10^−4^	1.00 × 10^−2^	COX6A1	cytochrome c oxidase subunit 6A2, mitochondrial−like
Unigene0042173	−13.8188	6.69 × 10^−4^	7.43 × 10^−3^	DCXR	diacetyl/L−xylulose reductase
Unigene0026858	−13.5407	6.09 × 10^−4^	6.85 × 10^−3^	−−	integumentary mucin C.1
Unigene0023802	−13.4138	1.53 × 10^−3^	1.52 × 10^−2^	SUMO3	small ubiquitin−related modifier 3−like
Unigene0035359	−11.1248	4.48 × 10^−10^	1.43 × 10^−8^	−−	−
Unigene0049053	−10.5651	7.65 × 10^−5^	1.06 × 10^−3^	cpr−5	putative cathepsin precursor
Unigene0010013	−10.5119	2.88 × 10^−3^	2.62 × 10^−2^	−−	uncharacterized protein LOC100158873 precursor
Unigene0054281	−9.4089	5.05 × 10^−4^	5.79 × 10^−3^	−−	uncharacterized LOC100166220

Description is determined by BLASTX. Fold change is calculated by RPKM. FDR, false discovery rate.

**Table 3 plants-11-03352-t003:** List of identified proteins and related metabolic pathways in aphids.

Spot No.	Average Normalized Volume		NCBI Accession	Protein Identification	Source	Mascot Score	MS Coverage	Peptide No.	MW	pI−Value
	STY	HDZ								
Amino acid metabolism										
1206	1.488	1.053	gi|1140522677	phosphoserine aminotransferase 1	*Bombyx mori*	76	24	9/65	40,345	6.97
1110	1.433	0.81	gi|193700145	aldehyde dehydrogenase, mitochondrial−like isoform 2	*Acyrthosiphon pisum*	49	13	5/17	52,342	6.96
1224	1.56	0.646	gi|399763011	phospholipid hydroperoxide glutathione peroxidase 1	*Chironomus riparius*	75	32	7/38	22,619	9.5
2108	0.967	1.237	gi|332018375	serine/threonine−protein phosphatase 2A regulatory subunit B″ subunit alpha	*Acromyrmex echinatior*	78	11	13/53	140,061	6.5
2701	0.758	1.034	gi|156541542	isochorismatase domain−containing protein 2, mitochondrial−like	*Nasonia vitripennis*	72	21	5/31	22,882	9.3
Bacterial metabolism										
2996	1.378	0.979	gi|285430	symbionin symL	*Acyrthosiphon pisum*	79	25	13/67	57,989	4.9
943	1.435	0.461	gi|285430	symbionin symL	*Acyrthosiphon pisum*	82	38	13/60	57,989	4.9
Carbohydrate metabolism										
1956	1.358	0.98	gi|193666869	isocitrate dehydrogenase [NADP] cytoplasmic−like	*Acyrthosiphon pisum*	55	24	9/78	46,850	6.19
1398	1.432	1.017	gi|52630947	putative fructose 1,6−bisphosphate aldolase	*Toxoptera citricida*	110	39	10/56	40,275	6.62
1017	1.159	0.848	gi|189240668	glucosyl/glucuronosyl transferases	*Tribolium castaneum*	64	12	4/6	52,338	9.2
2965	1.319	0.779	gi|215510634	endothelin−converting enzyme, putative	*Ixodes scapularis*	57	42	6/42	21,757	8.85
1410	1.606	0.938	gi|48096138	sorbitol dehydrogenase−like isoform 2	*Apis mellifera*	53	22	8/39	38,575	6.71
1226	1.377	0.759	gi|328699665	enolase−like isoform 2	*Acyrthosiphon pisum*	90	38	13/67	52,319	6.07
767	1.323	0.871	gi|301072331	beta−1,3−galactosyltransferase	*Helicoverpa armigera*	72	22	9/53	41,275	8.27
1646	1.823	0.781	gi|328722668	pyruvate dehydrogenase phosphatase regulatory subunit, mitochondrial−like isoform 1	*Acyrthosiphon pisum*	51	12	10/57	101,933	7.28
2521	0.741	1.336	gi|24647881	malate dehydrogenase 2	*Drosophila melanogaster*	71	31	9/47	35,317	9.2
658	0.909	1.497	gi|157128270	alpha−1,3−mannosyl−glycoprotein beta−1, 2−*N*−acetylglucosaminyltransferase	*Aedes aegypti*	59	19	9/59	54,409	8.85
Cell signaling										
2198	1.344	0.905	gi|193613348	rho GTPase−activating protein 17−like	*Acyrthosiphon pisum*	52	13	9/55	84,056	6.65
2154	1.266	0.91	gi|244790059	proteasome beta 2 subunit	*Acyrthosiphon pisum*	48	32	6/63	24,046	6.9
1814	0.909	0.665	gi|157128593	proteasome subunit beta type	*Aedes aegypti*	62	29	5/20	23,145	6.16
1201	1.487	0.935	gi|54287934	26S protease regulatory subunit−like protein	*Toxoptera citricida*	123	38	15/57	49,404	5.35
1184	1.433	0.93	gi|193617698	26S protease regulatory subunit 4−like	*Acyrthosiphon pisum*	102	25	12/34	49,426	6.23
570	1.171	0.485	gi|328712300	cyclin A2	*Acyrthosiphon pisum*	65	30	10/85	53,444	6.81
956	1.817	0.494	gi|345495296	nesprin−1−like	*Nasonia vitripennis*	65	10	32/63	446,115	5.51
850	1.003	1.331	gi|328724785	multidrug resistance−associated protein lethal (2)03659−like	*Acyrthosiphon pisum*	71	11	13/41	142,857	6.04
1435	1.138	0.998	gi|328707384	photoreceptor−specific nuclear receptor−like	*Acyrthosiphon pisum*	62	17	5/11	56,201	8.11
Cytoskeleton										
999	1.238	0.533	gi|240849384	roadblock−like	*Acyrthosiphon pisum*	55	23	10/56	11,179	6.06
1074	1.414	0.737	gi|298676439	tubulin beta−1	*Acyrthosiphon pisum*	94	31	16/86	50,637	4.72
3003	1.705	0.827	gi|193594183	tubulin alpha chain−like	*Acyrthosiphon pisum*	107	45	16/85	50,550	5.01
1385	1.842	0.822	gi|217330650	actin related protein 1	*Acyrthosiphon pisum*	123	47	16/67	42,158	5.29
1271	1.2	1.501	gi|298676439	tubulin beta−1	*Acyrthosiphon pisum*	82	33	12/57	50,363	4.79
3032	1.147	1.518	gi|193681197	actin−87E−like	*Acyrthosiphon pisum*	97	36	9/38	31,104	5.36
777	1.126	0.864	gi|512918251	cytospin−A−like	*Bombyx mori*	64	7	8/33	97,864	5.51
Energy metabolism										
1728	1.475	0.994	gi|350404548	ATP synthase subunit alpha, mitochondrial−like	*Acyrthosiphon pisum*	82	19	10/38	59,986	9.7
927	1.382	0.732	gi|328708451	PREDICTED: 4−coumarate−−CoA ligase 3−like	*Acyrthosiphon pisum*	97	17	9/39	67,319	8.8
765	1.191	0.647	gi|340723844	peroxisomal membrane protein PEX14−like	*Bombus terrestris*	53	29	8/57	30,199	6.03
3030	1.636	0.899	gi|328717825	peroxisomal acyl−coenzyme A oxidase 1−like	*Acyrthosiphon pisum*	63	14	5/9	76,327	5.99
1120	1.485	0.769	gi|209915626	ATP synthase subunit beta, mitochondrial	*Acyrthosiphon pisum*	180	54	18/54	55,777	4.9
1092	1.583	0.758	gi|328716950	PREDICTED: v−type proton ATPase subunit B−like	*Acyrthosiphon pisum*	98	27	11/64	55,565	5.44
3022	1.623	0.682	gi|328716950	PREDICTED: v−type proton ATPase subunit B−like	*Acyrthosiphon pisum*	110	31	13/42	55,565	5.3
2009	1.273	1.656	gi|209915626	ATP synthase subunit beta, mitochondrial	*Acyrthosiphon pisum*	92	38	11/76	37,568	4.96
Genetic information										
3024	1.018	0.646	gi|193667016	replication protein A 70 kDa DNA−binding subunit−like	*Acyrthosiphon pisum*	65	16	9/29	67,987	5.78
1273	1.407	0.94	gi|193664366	eukaryotic initiation factor 4A−like	*Acyrthosiphon pisum*	160	42	19/64	46,989	5.3
1202	1.507	0.965	gi|328712346	lysyl−tRNA synthetase−like isoform 2	*Acyrthosiphon pisum*	53	16	9/45	66,626	6.01
1506	1.107	0.677	gi|244790117	spindle and KT−associated 1	*Acyrthosiphon pisum*	56	39	4/51	33,646	6.11
957	1.664	0.974	gi|157118927	DEAD box ATP−dependent RNA helicase	*Aedes aegypti*	80	20	16/53	88,423	9.56
1636	1.478	0.619	gi|14531541	reverse transcriptase	*Chironomus sp.* ‘*februarius*’	70	46	6/40	17,516	9.67
846	1.55	0.568	gi|328719935	DNA ligase 1−like	*Acyrthosiphon pisum*	53	13	14/72	105,251	8.57
2963	1.572	0.5	gi|193702215	nuclear pore complex protein Nup50−like	*Acyrthosiphon pisum*	58	19	10/56	56,854	9.2
2575	0.814	1.539	gi|332022403	mariner Mos1 transposase	*Acromyrmex echinatior*	70	39	5/35	5489	10.1
2441	0.723	1.13	gi|170035055	cell cycle checkpoint protein rad17	*Culex quinquefasciatus*	83	21	9/26	58,917	9.19
2227	1.056	1.302	gi|332029719	DNA repair protein complementing XP−G cells−like protein	*Acromyrmex echinatior*	52	11	6/16	48,472	9.9
Membrane transport										
1798	1.208	0.915	gi|242247625	nascent polypeptide−associated complex subunit alpha	*Acyrthosiphon pisum*	43	7	27	22,784	4.8
1126	1.204	0.821	gi|328708774	SEC7 domain−containing protein 1−like	*Acyrthosiphon pisum*	49	7	5/10	93,617	9.03
1607	1.209	0.689	gi|114052995	Erg28−domain containing protein	*Bombyx mori*	70	15	4/6	20,146	9.89
937	1.39	0.75	gi|498925934	alpha−tocopherol transfer protein−like isoform X1	*Ceratitis capitata*	75	38	9/63	35,307	8.85
1231	1.591	1.226	gi|328699660	huntingtin−interacting protein 1−like isoform 2	*Acyrthosiphon pisum*	62	14	16/65	152,986	5.56
Nucleotide metabolism										
2959	1.461	0.939	gi|193669445	enolase−like isoform 1	*Acyrthosiphon pisum*	46	19	7/45	47,492	5.59
1093	1.294	0.771	gi|328700737	helicase SKI2W−like	*Acyrthosiphon pisum*	47	4	6/12	136,902	5.83
Protein synthesis										
852	1.319	0.782	gi|193652519	dnaJ homolog subfamily C member 8−like	*Acyrthosiphon pisum*	69	35	7/68	30,118	9.13
775	1.382	0.775	gi|193713655	protein disulfide−isomerase A3−like	*Acyrthosiphon pisum*	87	57	10/75	21,437	4.86
2975	1.15	0.65	gi|19365748	zinc finger protein 512B−like	*Acyrthosiphon pisum*	50	15	8/40	31,684	8.54
1177	1.158	0.661	gi|193656973	protein disulfide−isomerase−like	*Acyrthosiphon pisum*	73	15	9/50	57,489	4.7
1005	1.671	0.857	gi|193577789	t−complex protein 1 subunit eta−like	*Acyrthosiphon pisum*	91	22	11/42	59,872	6.55
3034	1.663	0.761	gi|193676235	t−complex protein 1 subunit theta−like	*Acyrthosiphon pisum*	83	25	12/48	60,325	5.2
1180	1.107	0.503	gi|193713655	protein disulfide−isomerase A3−like	*Acyrthosiphon pisum*	59	21	8/38	55,623	5.45
435	1.617	0.723	gi|193617621	transitional endoplasmic reticulum ATPase TER94−like	*Acyrthosiphon pisum*	102	26	20/56	89,914	5.1
785	1.868	0.781	gi|240848725	protein karl precursor	*Acyrthosiphon pisum*	51	15	5/34	28,596	5.64
472	1.389	0.459	gi|328725461	hcp beta−lactamase−like protein CG13865−like	*Acyrthosiphon pisum*	71	25	7/30	26,554	5.79
421	1.61	0.484	gi|193690671	elongation factor 2−like	*Acyrthosiphon pisum*	111	18	16/34	95,558	6.03
1954	0.701	1.193	gi|6856270	elongation factor−1 alpha	*Tylocentrus reticulatus*	61	20	6/23	33,614	8.8
2329	0.848	1.355	gi|229577161	GTPase 1 homolog	*Acyrthosiphon pisum*	75	26	8/30	68,643	5.52
739	1.095	0.976	gi|242397408	heat shock protein cognate 3 precursor	*Acyrthosiphon pisum*	123	34	19/69	72,993	5.1
1679	1.12	0.938	gi|193618024	116 kDa U5 small nuclear ribonucleoprotein component−like isoform 3	*Acyrthosiphon pisum*	75	16	14/61	109,788	5.01
1758	1.084	0.945	gi|328700367	28S ribosomal protein S5, mitochondrial−like	*Acyrthosiphon pisum*	73	25	11/51	47,201	9.87
2028	1.068	0.903	gi|33518699	antigen−5−like protein precursor	*Rhodnius prolixus*	65	36	5/28	28,228	9.12
2637	0.832	0.65	gi|112982956	splicing factor arginine/serine−rich 6	*Bombyx mori*	75	23	6/35	35,506	11.37
Signaling pathway										
650	0.846	0.548	gi|212505341	translational activator GCN1, putative	*Pediculus humanus corporis*	81	10	20/48	294,378	8.5
663	1.156	0.661	gi|345488865	ras−specific guanine nucleotide−releasing factor 1	*Nasonia vitripennis*	63	7	9/21	172,618	7.9
860	1.162	0.58	gi|240848707	protein enhancer of sevenless 2B−like	*Acyrthosiphon pisum*	76	58	11/85	25,139	5.31
Stress response										
907	1.421	0.932	gi|193603576	heat shock 70 kDa protein cognate 4−like isoform 2	*Acyrthosiphon pisum*	96	27	16/56	71,626	5.2
2972	1.317	0.813	gi|193603576	PREDICTED: heat shock 70 kDa protein cognate 4−like isoform 2	*Acyrthosiphon pisum*	149	46	22/73	71,626	5.34
766	1.388	0.793	gi|398025479	heat shock protein 70	*Aphis glycines*	114	16	17/49	71,399	5.3
3019	1.548	0.702	gi|193652748	heat shock protein 83−like	*Acyrthosiphon pisum*	127	28	21/70	83,707	4.8
1358	0.979	1.195	gi|193652748	heat shock protein 83−like	*Acyrthosiphon pisum*	58	12	9/28	83,707	4.8
Terpenoid backbone biosynthesis										
844	1.092	0.804	gi|240849357	dehydrodolichyl diphosphate synthase−like	*Acyrthosiphon pisum*	59	28	16/76	35,062	6.52

Numbers in cells correspond to the spot number on the 2D−DIGE gel. Red represents the downregulated proteins and green represents the upregulated ones of *Sitobion miscanthi*. The darker the color, the greater the change in protein expression (1− to 5−fold ratio for both geographic populations).

## Data Availability

All available data are contained within the article.

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
