# Peer review of "Proteomic and Transcriptomic Analysis for Identification of Endosymbiotic Bacteria Associated with BYDV Transmission Efficiency by *Sitobion miscanthi"

_plants, 2022, doi:10.3390/plants11233352_

Round 1
Reviewer 1 Report
Endosymbionts can have positive and negative effects on the ability of aphids to transmit plant viruses and it is important to characterise these effects which can assist in determining disease threat from infestations and also provide pointers for altering the microbiome of aphids for reducing the threat of virus transfer. In this paper the authors show quite strong albeit background-dependent effects of antibiotics affecting endosymbionts on virus transmission potential so this paper adds knowledge relevant to this issue.
Abstract
Authors indicate % reduction in transmission efficiency which is important but it would be better to have average changes rather than changes “by as 22 much as” since averages indicate overall effects.
Introduction
47-48 This is an odd sentence because the second part of the sentence is already implied by the first part. Reword.
61 Not clear which “lineages” are being referred to here.
63 Is this true? Buchnera may well be required by many aphids.
65 “Assuming” is the wrong word here, this is known.
Last paragraph. There are errors here, it needs to be read by a native English speaker. Also I would have thought that the transcription/proteome work would require more justification, it seems like an add on at the moment.
Methods
318-320 This does not make a lot of sense. If different individuals were collected, I assume they were nevertheless the same clone. How was this determined genetically?
321 So to follow on, were the clonal lines established from a single female or multiple field individuals?
327 Separate from what?
342 More information is needed on the nature of replication in the experiments. What constitutes a replicate?
362 Something missing after “symbiotic”. Methods generally need a careful proofreading.
386 How were data normalised?
480-484 I don’t really understand what is being compared here, re-express.
Results
Table 1 seems impossible to follow. There is one t value marked ** that is not significant, it is hard to separate actual transmission from “transmission efficiency”, and the table lacks values that allow for a direct comparison between controls and treatment values. I’d restructure the table with these points in mind. Note that some information could be left to a note such as the plant numbers with virus. The table should allow an easy comparison among treatments which is currently not possible. Error values should also be defined in table headings.
Table 2 is also hard to follow. Leave out the endosymbionts that were not detected at all. Define all terms (including PASS etc). Also I would rather see quantitative values in the table to see if Buchnera density has decreased. I realize this is plotted in Fig 1 but it would be good to have values and SE in the table as well.
NOTE that the secondary endosymbionts detected should also be mentioned in the abstract somewhere and the impact of the antibiotics on them.
NOTE I suspect that the labelling on Fig 1 should be improved. The x axis should just indicate treatments, not the endosymbionts tested which can appear in a separate label below.
NOTE Although this is implied elsewhere it is important for the authors to reiterate the treatments in the results – so are these antibiotic effects immediately apparent after treatments or is the test being done on the next generation of aphids subsequent to the treatments being applied. I kept pondering this given that endosymbiont removal can be difficult to separate from other effects of antibiotics.
The transcription-proteome work looks OK but I found it hard to interpret because it was unclear to me how many of the changes were due to (a) the loss of endosymbionts, (b) antibiotic related effects not connected to endosymbiont changes, and (3) virus interactions. It is a shame that the authors did not also compare lines a generation later when presumably some endosymbionts would still remain eradicated but there are fewer confounding effects, particularly if virus infested and virus free aphids were part of the comparison.
Discussion
192-194 So why did you think the results here were so different to previous work in terms of elimination? Was your Buchnera assay more sensitive?
204 This expectation is based on the nature of the endosymbionts present. For instance Regiella apparently decreases virus transmission.
220-226 I don’t really understand how you can separate Buchnera density effects from Rickettsia effects.
There is a lot of speculation about possible virus-protein interactions but it is really hard to tell if these mean much in the absence of additional treatments and finer level studies. Discussion could use a concluding paragraph.
Reviewer 2 Report
The authors in the article entitled, ‘Proteomic and transcriptomic analysis for Identification of endosymbiotic bacteria associated with BYDV transmission efficiency by Sitobion miscanthi’ aimed at investigating the changes in the composition of the endosymbiont population affected BYDV 21 transmission efficiency in two aphid clones. In my opinion, this research is relevant and will contribute to the knowledge base of the control of aphids. However, l also believe that the current manuscript can be edified to convey intended information by the authors. Below are listed some of my suggestions:
Q1. The whole manuscript requires English revamp in grammatical expressions and academic content delivery. For instance, Line 18 (…but little is known the interactions between endosymbionts….), Line 318 (…contrary to HDZ WHO had very low efficiency..), who is a pronoun for animate subjects, …. etc.
Q2. The Introduction lacks background study from previous studies to support the intend of this research.
Q3. In result section, the authors put more effort in graphical presentation of their findings. I suggest the authors to broaden the summary of the results, and explain their findings.
Q4. In text referencing, the authors may revise intext referencing according to the author guidelines of the Journal for Line 207, 215, 219, 249. For example, Sakura et al., [15].
Q5. In the Discussion section, section 3.1 can be broadened and more supporting work can be added.
Q6. The Materials and Method section is lengthier, the authors may paraphrase this section highlighting important details. For example, the Statistical Analyses for both experiments can be paraphrased under a single sub-section.
Q7. The authors can also give a detailed conclusion of the findings.
Q8. Figures in the manuscript can be enhanced, and the resolution increased.
Q9. The authors cited many old references, I suggest the authors to include most recent citations similar to this research.
Round 2
Reviewer 1 Report
There are still issues with this paper.
79 Last sentence of the Introduction still does not make sense. I don’t follow “affected by…. and proteins” because you are looking at effects on proteins.
Methods
318-320 This does not make a lot of sense. If different individuals were collected, I assume they were nevertheless the same clone. How was this determined genetically?
I don’t see where this comment is addressed. Also the sentence at 417-420 is impossible to follow.
321 So to follow on, were the clonal lines established from a single female or multiple field individuals?
I don’t see this comment being addressed.
342 More information is needed on the nature of replication in the experiments. What constitutes a replicate?
I still don’t know what a “wheat replicate” is. Assume each replicate is an individual plant? Also on line 466 the authors should indicate “inhibition rate” not “inhibit rate”.
Results
Table 1 still needs improvement. P=0.0001 should presumably be P<0.0001. Two percentages are given in the brackets without being defined in the headings.
Table 2 suggestions I made were ignored.
Discussion
This is still hard to follow because of poor language.
